# Molecular Biology of Spermatogenesis: Novel Targets of Apparently Idiopathic Male Infertility

**DOI:** 10.3390/ijms21051728

**Published:** 2020-03-03

**Authors:** Rossella Cannarella, Rosita A. Condorelli, Laura M. Mongioì, Sandro La Vignera, Aldo E. Calogero

**Affiliations:** Department of Clinical and Experimental Medicine, University of Catania, 95123 Catania, Italy; rosita.condorelli@unict.it (R.A.C.); lauramongioi@unict.it (L.M.M.); acaloger@unict.it (A.E.C.)

**Keywords:** spermatogenetic failure, embryo growth, male infertility, spermatogenesis, recurrent pregnancy loss, sperm proteome, DNA fragmentation, sperm transcriptome

## Abstract

Male infertility affects half of infertile couples and, currently, a relevant percentage of cases of male infertility is considered as idiopathic. Although the male contribution to human fertilization has traditionally been restricted to sperm DNA, current evidence suggest that a relevant number of sperm transcripts and proteins are involved in acrosome reactions, sperm‒oocyte fusion and, once released into the oocyte, embryo growth and development. The aim of this review is to provide updated and comprehensive insight into the molecular biology of spermatogenesis, including evidence on spermatogenetic failure and underlining the role of the sperm-carried molecular factors involved in oocyte fertilization and embryo growth. This represents the first step in the identification of new possible diagnostic and, possibly, therapeutic markers in the field of apparently idiopathic male infertility.

## 1. Introduction

Infertility is a widespread condition in industrialized countries, affecting up to 15% of couples of childbearing age [1]. It is defined as the inability to achieve conception after 1–2 years of unprotected sexual intercourse [2]. Infertility results from an impairment of male and/or female gamete production, or the gametes’ inability to reach each other or fuse. In addition, once the embryo is generated, infertility may arise from insufficient embryo growth and development [3]. Several etiological factors have been identified in infertile couples thus far. These include (but they are not limited to) disorders of gametogenesis, abnormal gamete quality, genital tract dysfunctions (e.g., obstruction, infections, inflammations, etc.), abnormal embryo implant, or vascularization [3]. However, the etiology of infertility still remains unclear in a relevant percentage of cases [3].

Male factor infertility accounts, cumulatively, for half of couple infertility, being solely identified in 30% of couples [4]. Notably, a careful diagnostic and therapeutic approach to the male partner is often overlooked in the management of couple infertility, including couples with recurrent pregnancy loss (RPL) [3], since the overall contribution of the male gamete to embryogenesis has been thought to be restricted to the sperm DNA. Therefore, assisted reproductive technique (ART), primarily aimed at injecting the sperm DNA into the oocyte, has long been considered the most effective therapeutic approach to infertility, thus neglecting the diagnosis and treatment of the male partners in infertile couples. However, the high discrepancy between the number of ART cycles initiated and the number of clinical pregnancies achieved suggests that ART is not so powerful and other factors related to sperm quality, beyond the sperm DNA fragmentation, have to be taken into account. In addition, and worryingly, the evidence has shown that no etiological factor can be found in ~70% of the male partners of infertile couples and the cause of male infertility remains elusive [5,6]. In light of such data, further molecular targets of male infertility have to be investigated.

Spermatogenesis is a complex series of molecular events resulting in the proliferation of spermatogonial stem cells (SSCs) and in their differentiation into highly specialized, terminally differentiated spermatozoa. The development of the current high-throughput “omic” technologies has led to the recognition of the great complexity of the spermatozoon, which carries thousands of RNAs and proteins. The overall contribution of the sperm genome (including epigenetic regulation), transcriptome, and proteome to embryo formation and development needs to be investigated to identify novel molecular targets responsible for male infertility.

Based on these premises, the purpose of this review is to provide updated and comprehensive insights into the molecular biology of spermatogenesis, underlining the role of the molecular factors involved in oocyte fertilization and embryo growth that are produced during spermatogenesis and carried out by the spermatozoon.

## 2. Physiology of Spermatogenesis

Spermatogenesis, a 74-day-long process required to finally differentiate SSCs in spermatozoa, typically involves three different functional phases occurring in testicular seminiferous tubules: mitosis, meiosis, and spermiogenesis. A highly complex series of molecular events, requiring proper interactions between Sertoli cells, germ cells, epithelial tubular cells, and the integrity of the blood‒testis‒barrier (BTB), is needed for a successful spermatogenesis [7]. SSCs are undifferentiated spermatogonia capable of both self-renewal (to maintain the pool of stem cells) and differentiation into A-paired (Ap) spermatogonia. They later develop into A-aligned (Aal) spermatogonia by sequential mitotic divisions. SSCs, Ap, and Aal are undifferentiated spermatogonia and differ from differentiated ones in the expression of self-renewal and proliferating-associated genes (e.g., Grfα1, Ret, Nanos, Plzf, Id4, Pou5f1, Foxo1, Mir-17-92, Lin28a, Pax7, Neurog3, Sox3, Taf4b, Plap, Ap2y, and Sall4) [8,9].

Sertoli cell-derived, follicle stimulating hormone (FSH)-induced growth factors (e.g., the glial cell line-derived neurotrophic factor (GDNF)) boost Aal spermatogonia differentiation into A1 [8]. By mitotic divisions, they in turn transform into A2, A3, A4, and B spermatogonia. A1‒4 and B spermatogonia belong to the group of differentiated spermatogonia. These cells are characterized by a downregulation of self-renewal genes and an upregulation of those associated with differentiation (e.g., Sohlh1, Sohlh2, Dnmt1, c-kit, etc.) [9]. Subsequently, type B spermatogonia generate preleptotene spermatocytes, which require close contact with Sertoli cells and the integrity of the epithelium to be transported across the BTB to differentiate into leptotene, zygotene, and pachytene spermatocytes and enter the ad luminal compartment [7].

Preleptotene spermatocytes undergo a 16-day-long meiotic division to differentiate into secondary spermatocytes. During prophase I, the first step of meiosis, double-strand breaks (DSBs) (leptotene), pairing of homologue chromosome (zygotene), and crossing over (CO) (pachytene) take place, all relying on fine molecular mechanisms. Accordingly, several proteins with enzymatic activity are expressed in this step (such as PLK-4) to ensure proper chromosome segregation, but also SPO11β, RPA1, RPA2, RPA3, MEIOB, RAD51, SPATA22, MSH4, MSH5, TEX11, TEX15, SYCE, HSF2, HEI10, RNF212, and CNTD1, required for proper CO recombination and DSBs repair [9].

After CO has been completed, chromosome segregation (consisting of the homologues chromosome split up) occurs. It requires the presence of intact intercellular bridges, which are structures important for germ cell communication, synchronization, and chromosome dosage compensation in haploid cells. TEX14 is a structural component of intercellular bridges, highly conserved among mammals, which has been implicated in the pathogenesis of spermatogenetic failure (SPGF) [10].

Secondary spermatocytes undergo meiosis II, through which sister chromatids divide to produce haploid round spermatids. Later on, DNA packaging with protamines, acrosome and midpiece formation, flagella organization, and cytoplasm expulsion are required for the differentiation of secondary spermatocytes into spermatozoa. This process is called spermiogenesis and requires 26 days. Finally, differentiated spermatozoa are highly specialized cells that must be capable of active motility to get into the female genital tract and penetrate the oocyte. Due to the limited sperm scavenger capacity and sperm’s well-known susceptibility to reactive oxygen species (ROS)-induced damage, all environmental factors leading to a perturbation of the pro- and anti-oxidant balance can potentially interfere with the sperm fertilization capacity [11,12].

The molecular factors involved in spermatogenesis whose mutations have been called into play in the pathogenesis of SPGF are shown in Figure 1. In addition to these proteins, RNA processing appears to be involved in mammalian spermatogenesis, thus adding further complexity. In particular, post-transcriptional regulation of gene expression (e.g., pre-mRNA splicing, mRNA transport, decay or translation, and 3’-end processing) occurs by reversibly modifying mammalian mRNAs and lncRNAs through the N6-methylation of the adenosine residues (m6A), which is installed by a multicomponent methyltransferase complex made of METTL3, METTL4, WTAP, RBM15, and KIAA1429, and removed by the FTO and ALKBH5 proteins [13]. m6A RNA post-translational modifications have recently been reported to be essential for mouse spermatogenesis [13].

## 3. Physiology of Fertilization and Embryo Development

Fertilization is a multistep process starting once the spermatozoon has reached the oocyte. It requires the following events: capacitation, hyperactivation, acrosome reaction, binding of the spermatozoon to the zona pellucida (ZP), penetration of the ZP, fusion of the sperm with the oocyte plasma membrane, and oocyte activation (Figure 2).

An acrosome is a vesicle under the sperm membrane and above the nucleus, in the anterior part of the sperm head, that contains pivotal enzymes involved in sperm penetration into the oocyte [14,15]. The integrity of the inner acrosome membrane (IAM) (lying external to the nuclear membrane), the outer acrosome membrane (OAM) (placed below the plasma membrane), and the equatorial segment (ES) (the meeting point between the IAM and the OAM) is important for the acrosome reaction (AR) [16]. AR consists of the opening of 100 fusion pores between the OAM and the sperm membrane. Concomitantly, acrosome enzymes are released, leading the spermatozoon to penetrate the ZP; this process is favored by the sperm’s tail beating (hyperactivation). By this time, factors enhancing sperm penetration and motility are released by the cumulus cells [17,18]. These time-dependent physiological changes, known as capacitation, cannot occur prior to sperm binding to the ZP.

ZP is a highly specialized extracellular matrix surrounding the oocyte, containing three major glycoproteins: ZP1, ZP2, and ZP3. ZP1‒3 receptors are expressed in the sperm head membrane. Their binding to ZP proteins is crucial for the sperm‒oocyte recognition phase, which follows the AR [19]. After recognition, the oocyte‒sperm membranes fuse together and the oocyte is penetrated by the sperm head, centriole, midpiece, and principal piece [20]. Hence, the entire genome, transcriptome, and proteome carried by the spermatozoon are carried into the oocyte.

After sperm penetration into the oocyte, male and female pronuclei lie in the same cytoplasm and a series of events happen to make them fuse together. In particular, chromatin remodeling (involving histone incorporation), genome-wide DNA demethylation and remethylation, histone modification, establishment of open chromatin patterns, and changes in chromatin conformation take place in parental nuclei [21]. Sperm protamines, which keep the sperm DNA highly condensed up until fertilization, are replaced by maternally-derived histones [21]. Moreover, the paternal genome is subjected to global demethylation within the first 24 h of embryo development, much faster than the maternal genome [22]. However, the evidence has shown that parental imprinted regions escape from demethylation, although the reason for this is not fully understood [23]. Some researchers hypothesized that paternally-expressed imprinted genes (e.g., IGF2, a sperm-carried transcript [24]) may be of particular importance in preimplantation embryo development [25]. Accordingly, the involvement of imprinted genes in embryo and placenta development and growth is well acknowledged in mice. Commonly, paternally-imprinted genes are believed to act as embryo/fetal-growth promoters, while, conversely, maternally-imprinted ones oppose the embryo‘s growth [26]. Transcription is first detectable in the male pronucleus and, in humans, a transcription burst happens between the four-cell and eight-cell embryo stages, when the embryo gene activation (EGA) starts [15]. Hence, before EGA, preimplantation embryo development is only sustained by the sperm and oocyte-derived transcriptome and proteome, whose qualitative or quantitative alteration might cause embryonic development arrest. 

The fusion of male and female pronuclei forms the zygote. It will develop into a morula and then a blastocyst in about six days. The disappearance of the ZP surrounding the blastocyst follows implantation into the uterine wall, leading to the formation of ectoderm, endoderm, and mesoderm layers, needed to make a competent, functional, and independent organism [21].

## 4. Sperm’s Contribution to Human Fertility

Human spermatozoa contribute to many fundamental moments for human fertility (e.g., AR, ZP binding and penetration, sperm‒oocyte fusion, and embryo growth and development) through their genome, transcriptome, and proteome, whose products are transported inside the oocyte at the time of fertilization.

### 4.1. Sperm Genome

A wide range of somatic gene variations has recently been called into play as possible causes of apparently unexplainable spermatogenetic failure. Mainly, these are genes involved in pivotal steps of spermatogenesis, such as germ cell proliferation, meiosis, and spermiogenesis (Table 1). Their exploration may increase the diagnostic yield in patients with an apparently idiopathic abnormal sperm count, motility, and/or morphology [6,9].

Beyond somatic gene abnormalities, the quality of the sperm genome is of great importance for the pregnancy outcome. The sperm genome is highly packaged by protamines, which replace histones during spermatogenesis. This process, which is typical of spermatozoa, is called protamination and ensures significant DNA compaction, resulting in protection from external insults and gene silencing [25]. Abnormal sperm protamination represents one of the mechanisms by which sperm DNA integrity can be affected, with consequent defective development of the embryo. Indeed, although the oocyte is able to repair the sperm DNA damage that occurs during their passage in both the male and female genital tracts, an excessive degree of damage that exceeds the repairing ability of the oocyte will result in an adverse pregnancy outcome [27]. The negative impact of sperm DNA fragmentation on the pregnancy outcome (including recurrent pregnancy loss) has already been confirmed by three different meta-analyses [27,28,29]. In line with this evidence, abnormal sperm protamination (e.g., altered P1/P2 ratio) is associated with poor sperm and embryo quality, as well as with low fertilization and pregnancy rates [30].

Protamination involves ~85% of sperm DNA, whereas the other ~15% remains bound to histones. This amount of histone-bound DNA is not randomly distributed, but is placed in sites playing a structural role (e.g., telomeric and peri-centromeric regions) or in promoter regions of genes involved in embryo development [31]. Post-translational histone modifications consist of methylation, acetylation, and phosphorylation of the histones’ N-terminal domain. These changes modify the chromatin electrostatic charge and, therefore, the structural organization and DNA accessibility to transcription factors [31]. Thus, these regions may be susceptible to external damage in the absence of a specific sperm compactness abnormality.

The exposure to pro-oxidant factors can concomitantly damage both sperm DNA integrity and gene expression by inducing epigenetic modifications. Accordingly, ROS have been found capable of altering H19‒Igf2 methylation (by reducing H19 and increasing Igf2 methylation pattern) [32]. Moreover, abnormal methylation of the imprinted genes IGF2 and KCNQ1 at the spermatozoon level has recently been reported in infertile patients, in association with an increased sperm DNA fragmentation [33]. Furthermore, global sperm DNA methylation has been shown to inversely correlate with sperm DNA fragmentation [34]. In line with these findings, meta-analytic data report a decrease in the sperm H19‒IGF2 methylation rate in infertile patients compared to fertile controls [35]. Importantly, methylation of imprinted genes is inherited by the embryo [26] and, therefore, this abnormality can be transmitted to the offspring.

Finally, in addition to sperm DNA fragmentation (whose role is well recognized), sperm global methylation or methylation of imprinted genes (Table 2) might also represent good candidates to be taken into account in apparently idiopathic infertile patients in the attempt to find an etiologic factor.

### 4.2. Sperm Transcriptome

Mature spermatozoa have generally been seen as a simple carrier of the paternal genome. Indeed, due to the chromatin compactness, it was believed that these cells were totally incapable of gene transcription and translation. However, some hypo-methylated DNA regions (e.g., where the paternally-expressed imprinted genes map) allow access to the transcription apparatus. Furthermore, despite the majority of sperm RNA probably being transcribed prior meiosis, some evidence has shown that transcription occurs in postmeiotic spermatozoa [36,37]. Since the amount of cellular energy has to be spared by the spermatozoon to face the energy-consuming processes involved in fertilization, postmeiotic RNA transcription is likely required for sperm fertilization or, once released in the oocyte, for embryo development.

The involvement of sperm transcriptome in the pregnancy outcome has been shown in mice. Accordingly, the cleavage of sperm RNAs by treatment with a RNAse drastically decreased the rate of blastocyst formation and the live birth rate of embryos produced by intracytoplasmic sperm injection (ICSI) compared to those obtained with untreated spermatozoa [38]. In addition, the seminal fluid transcriptome also seems to play a role in embryo development. In particular, animal studies have shown a lack of embryo growth after the removal of sperm epididymal miRNAs [39]. Moreover, a different sperm transcriptome profile has been reported between samples able to achieve pregnancy compared with those that were not able to do so [40]. This further confirms the involvement of the sperm transcriptome in human fertility.

Spermatozoa carry thousands of different RNAs—such as messenger RNA (mRNA), micro-RNA (miRNA), interference RNA (iRNA), antisense RNA and long non-coding RNA—and more than 4000 different mRNAs [25]. Human sperm transcription of the paternally-derived imprinted gene IGF2 has been demonstrated [24]. Imprinting has been extensively studied in humans and mice. The role of imprinted genes in processes involved in embryo and placenta development and growth is well recognized in mice. Given that the Igf2 murine gene encodes for a growth factor promoting fetal and placental growth [26], the amount of the sperm-derived IGF2 transcript released into the human oocyte after sperm‒oocyte fusion may influence the embryo morphokinetics. Accordingly, as previously summarized, preimplantation embryo development is only sustained by the sperm and oocyte-derived transcriptomes and proteome before EGA, since the human embryo transcription burst occurs only between the four-cell and eight-cell embryo stages [21].

IGF2 is the first historically characterized imprinted gene whose expression depends on H19 methylation. While in the maternal allele H19 unmethylation prevents the accession to the IGF2 enhancer, H19 methylation leads to IGF2 expression in the paternal one [41]. Interestingly, H19 and IGF2 are expressed in the human placenta and their transcripts have recently been investigated in human placentas from ART and from natural conception. H19 resulted upregulated and IGF2 downregulated in placentas from ART compared to those from natural conception [42]. Whether sperm IGF2 downregulation represents a target for apparently idiopathic infertility remains to be established.

### 4.3. Sperm Proteome

Several studies have analyzed the sperm proteome; this has resulted in the characterization of 6871 proteins so far [22]. These proteins have been functionally analyzed using Gene Ontology (GO) annotations and Mouse Genome Informatics (MGI) databases. This brought about novel, interesting insights into the role of the sperm proteome in fertilization, pre-implantation embryo development, and paternal epigenetic inheritance [22]. Particularly, among 6871 proteins, a total of 103 have been predicted to be involved in capacitation, AR, ZP binding and penetration and sperm‒oocyte fusion (Table 3). These proteins include AKAP3, AKAP4, or CATSPERs, involved in AR, of IZUMO I and PCCZ I, playing a role in sperm‒oocyte fusion and oocyte activation, respectively, of PCSK4 and CRISP I, required for ZP binding [22]. Notably, 93 proteins have been functionally addressed as involved in fertility by biological evidence. In fact, beside the bio-informatic analysis, animal studies have confirmed the role of 59 sperm proteins in embryo development. Specifically, at the early embryo stage (from zygotes to eight-celled embryos), the spermatozoon delivers 11 lethality-related proteins that are not expressed by the oocyte, such as DSC3, which is needed for cell adhesion (a process required for blastomeric formation) [43]. Furthermore, 29 and 19 sperm-specific proteins have been characterized as involved in morula and blastocyst formation, respectively (e.g., B3GNT5 or PCYTIA) [22].

GO annotation also provided evidence for the presence of 560 sperm proteins that are able to modulate gene expression. They are subdivided into transcription factors (*n* = 381), chromatin remodeling proteins acting on DNA methylation (*n* = 25) or histone post-translational modification (*n* = 118), proteins involved in non-coding RNA processing (*n* = 36) [22]. Furthermore, 28 proteins might impact on epigenetic inheritance and early embryo gene expression [22]. Finally, some of the proteins predicted to be involved in fertilization are encoded by genes responsible for spermatogenetic failure (e.g., CATSPER1, TEX, and SYCP proteins) (Table 1 and Table 3), which deserve further investigation.

### 4.4. Sperm Metabolome

The metabolome includes all the metabolites, regardless of their biochemical nature, involved in biological processes. As several metabolic reactions are needed for human spermatogenesis to occur properly, the study of sperm metabolome (including seminal plasma) has captured the attention of scientists in recent years, since it may aid in the understanding of apparently idiopathic infertility.

Amino acids, biogenic amines, and lipids represent the majority of the sperm metabolome, and have been found to be related to conventional sperm parameters. In more detail, the nonessential amino acid glutamine (Gln), which has been reported to be the most abundant amino acid in both spermatozoa and seminal plasma, correlates significantly with sperm concentration and motility [44]. Arginine (Arg) was described as crucial for spermatogenesis and, as a consequence, Arg-deficient diets are associated with the presence of multinuclear giant cells in the testis and SPGF [45]. Interestingly, Arg supplementation improves the number of human motile spermatozoa [46,47]. In a similar manner, the sperm concentration of acyl-carnitines is positively associated with sperm motility [48].

In addition to amino acids, sperm lipid content has been associated with progressive sperm motility. Furthermore, the sperm concentration of total, HDL and LDL cholesterol, and triglycerides is higher in patients with oligo-astheno-teratozoospermia compared with normozoospermic men [49].

These findings may help to address forms of apparently idiopathic male infertility and to design targeted therapies in the future. Studies investigating differences in the sperm metabolome between fertile and infertile patients are warranted.

## 5. Discussion

Recent advantages in the field of sperm “omics” are leading to a new understanding of the role of spermatozoa in fertility. Consequently, based on the evidence reviewed in this article, spermatozoa are not only simple transporters of their DNA within the oocyte, but also carry a wide range of important molecules (e.g., transcripts, proteins, and metabolites) that are useful for supporting embryo mitosis.

We suggest that the role of paternally-expressed imprinted genes should not be underestimated since transcription may potentially occur in the spermatozoon. Accordingly, these genes display a low-methylated pattern in their promoter region, which is consistent with transcription. This hypothesis is supported by the expression of IGF2 mRNA into human spermatozoa [24], and the homonym protein plays a role in prenatal fetal development. As a growth factor, IGF2 may play a role in the cellular mitosis of the embryo [25]. Furthermore, since the expression of IGF2 depends on the promotor methylation rate, which the embryo receives from the spermatozoon, it may be speculated that sperm-expressed IGF2 plays a role in embryo growth at the very early stage of embryo formation before EGA. The embryonal expression of IGF2 (which depends on the promoter gene methylation rate of the embryo paternal chromosome) may influence embryo mitosis after EGA. If future studies confirm the role of sperm-carried IGF2 in human fertility, this may become a novel diagnostic and/or therapeutic target in the field of apparently idiopathic male infertility.

In addition to IGF2, several other sperm-carried factors have emerged from this “omic” analysis. In the near future, targeted study of the role of such molecules in human fertility will be needed to assess their levels in fertile and infertile patients—to be further supported by data collected in assisted reproductive techniques (ARTs) centers, prior to their becoming available (if supported by research findings) in clinical practice.

## 6. Conclusions

In conclusion, several lines of evidence support a role for human spermatozoa in fertilization that goes beyond its DNA. A number of transcripts and proteins are transported within the oocyte and impact on embryo growth and development. Their characterization will likely bring about the recognition of novel diagnostic and, possibly, therapeutic targets in what is currently considered idiopathic male infertility.

## Figures and Tables

**Figure 1 ijms-21-01728-f001:**
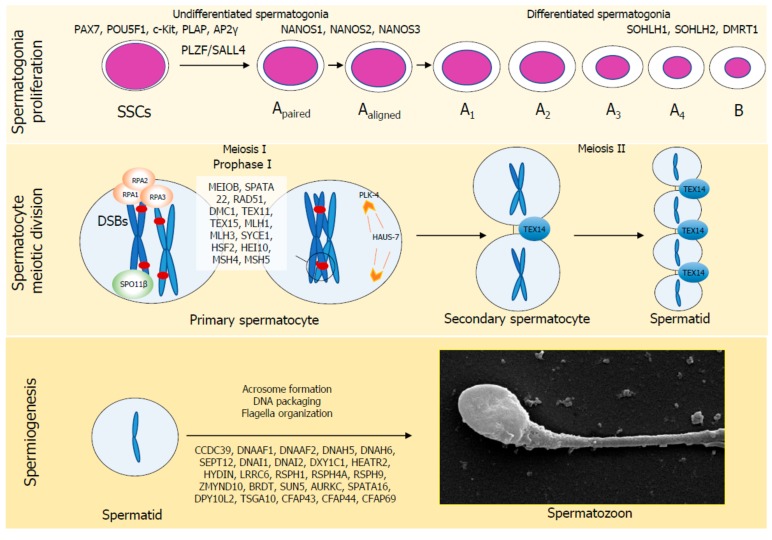
Molecular factors involved in human spermatogenesis. Spermatogenesis is a 74-day-long process required to differentiate spermatogonial stem cells in spermatozoa, divided in three functional steps: spermatogonia proliferation, spermatocyte meiotic division and spermiogenesis. Several factors are specifically involved in each of these steps. Their absence results in the failure of spermatogenesis in a specific phase.

**Figure 2 ijms-21-01728-f002:**
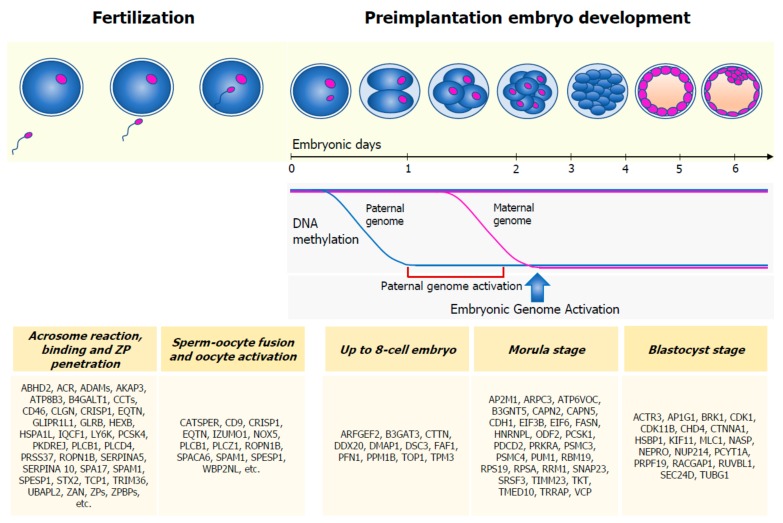
Sperm contribution in fertilization and preimplantation embryo development. Several sperm-carried proteins are involved in acrosome reaction, binding to the zona pellucida (ZP), sperm‒oocyte fusion, and oocyte activation. Soon after sperm penetration into the oocyte, the sperm genome undergoes to global demethylation, much faster than the maternal genome. Embryonic Genome Activation (EGA), a process by which the embryo starts to produce its own transcripts, starts between the four-cell and eight-cell embryo stages. Before this phase, the embryo is sustained by the gametes’ transcriptome and proteome. A list of sperm-specific proteins has been identified in the embryo, in the up to eight-cell, morula, and blastocyst phases.

**Table 1 ijms-21-01728-t001:** Genes involved in spermatogenetic failure.

Sperm Parameter	Genes
Sperm number	*CATSPER1, CCDC39, DAZ1, DAZ2, DAZ3, DAZ4, DBY, DMC1, DMRT1, DNAH6, FANCM, HAUS7, HSF2, KLHL10, MEIOB, NR5A1, PLK-4, SEPT12, SOHLH1, SPINK2, SYCE1, SYCP3, TAF4B, TDRD6, TEX11, TEX14, TEX15, USP26, ZMYND15*
Sperm motility	*AK7, CCDC39, CEP135, CFAP43, CFAP44, CFAP69, DNAAF1, DNAAF2, DNAAF3, DNAH1, DNAH5, DNAI1, DNAI2, DNAJB13, DYX1C1, FSIP2, HEATR2, HYDIN, LRRC6, PIH1D3,* *RSPH1, RSPH4A, RSPH9, SLC26A8, WDR66, ZMYND10*
Sperm morphology	*AURKC, BRDT, DPY19L2, SPATA16, SUN5, TSGA10*

**Legend:** AK7, Adenylate kinase 7; AURKC, Aurora kinase C; BRDT, Bromodomain, testis-specific; CATSPER1, Cation channel, sperm-associated, 1; CCDC39, Coiled-coil domain-containing protein 39; CEP135, Centrosomal protein, 135 KD; CFAP43, Cilia- and flagella- associated protein 43; CFAP44, Cilia- and flagella- associated protein 44; CFAP69, Cilia- and flagella- associated protein 69; DAZ1, Deleted in azoospermia 1; DAZ2, Deleted in azoospermia 2; DAZ3, Deleted in azoospermia 3; DAZ4, Deleted in azoospermia 4; DBY (DDX3Y), Dead/H Box 3, Y-linked; DMC1, Disrupted meiotic Cdna 1, yeast, homolog of; DMRT1, Doublesex- and MAB3- related transcription factor 1; DNAAF1, Dynein, axonemal, assembly factor 1; DNAAF2, Dynein, axonemal, assembly factor 2; DNAAF3, Dynein, axonemal, assembly factor 3; DNAH1, Dynein, axonemal, heavy chain 1; DNAH5, Dynein, axonemal, heavy chain 5; DNAH6, Dynein, axonemal, heavy chain 6; DNAI1, Dynein, axonemal, intermediate chain 1; DNAI2, Dynein, axonemal, intermediate chain 2; DNAJB13, DNAJ/HSP40 homolg, subfamily B, member 13; DPY19L2, DPY19-like 2; DYX1C1 (DNAAF4), Dynein axonemal assembly factor 4; FANCM, FANCM gene; HAUS7, Haus Augmin-like complex, subunit 7; HEATR2 (DNAAF5), Heat repeat-containing protein 2; HSF2, Heat shock-transcription factor 2; HYDIN, Hydrocephalus-inducing, mouse, homolog of; KLHL10, Kelch-like 10; LRRC6, Leucine-rich repeat-containing protein 6; MEIOB, Meiosis-specific protein with OB domains; NR5A1, Nuclear receptor subfamily 5, group A, member 1; PIH1D3, PIH1 domain-containing protein 3; PLK-4, Polo-like kinase 4; RSPH1, Radial spoke head 1, Chlamydomonas, homolog of; RSPH4A, Radial spoke head 4A, Chlamydomonas, homolog of; RSPH9, Radial spoke head 9, Chlamydomonas, homolog of; SEPT12, Septin 12; SLC26A8, Solute carrier family 26 (sulfate transporter), member 8; SOHLH1, Spermatogenesis- and oogenesis- specific basic helix-loop-helix protein 1; SPATA 16, Spermatogenesis-associated protein 16; SPINK2, Serine protease inhibitor, Kazal-type, 2; SUN5, SAD1 and UNC84 domain-containing protein 5; SYCE1, Synaptonemal complex central element protein 1; SYCP3, Synaptonemal complex protein 3; TAF4B, TAF4B RNA polymerase II, TATA Box-binding protein-associated factor; TDRD6, Tudor domain-containing protein 6; TEX11, Testis-expressed gene 11; TEX14, Testis-expressed gene 14; TEX15, Testis-expressed gene 15; TSGA10, Testis-specific protein 10; USP26, Ubiquitin-specific protease 26; WDR66, WD repeat-containing protein 66; ZMYND10, Zing finger mind-containing protein 10; ZMYND15, Zing finger mind-containing protein 15.

**Table 2 ijms-21-01728-t002:** Imprinted genes involved in placenta and embryo development and growth in mice and in human sperm quality.

Process	Imprinted Genes
***Imprinted genes involved in placenta and embryo development and growth in mice***	
Placenta establishing	*Peg10*
Nutrient transport capacity and surface area for exchange	*Igf2, Grb10*
Fetal growth	*Igf2, Igf2r, Cdkn1c, Grb10*
***Imprinted genes involved in sperm quality***	
*CREM*	Increased methylation is associated with decreased semen quality
*DAZL*	Increased methylation in OAT patients compared to controls
*FAM50B*	Decreased methylation levels are associated with asthenozoospermia
*GNAS*	Decreased methylation levels are associated with asthenozoospermia
*GLT2*	Abnormal methylation levels are associated with oligozoospermia
*H19*	Decreased methylation is associated with male infertility
*KCNQ1OT1*	Increased methylation in patients with abnormal sperm parameters
*MEST*	Increased methylation is associated with male infertility
*RHOX*	Increased methylation is associated with male infertility
*SNRPN*	Increased methylation is associated with male infertility
*ZAC*	Increased methylation is associated with oligozoospermia
*CREM*	Increased methylation is associated with decreased semen quality
*DAZL*	Increased methylation in OAT patients compared to controls
*FAM50B*	Decreased methylation levels are associated with asthenozoospermia

**Legend:***Cdkn1c*, Cyclin-dependent kinase inhibitor 1 c; *CREM***,** cAMP responsive element modulator; *DAZL*, Deleted in azoospermia-like; *FAM50B*, Family with sequence similarity 50, member B; *GNAS*, Guanidine nucleotide-binding protein, alpha-stimulating activity polypeptide 1; *GLT2*, Gene trap locus 2 (also known as *MEG3*, Maternally expressed gene 3); *Grb10*, growth factor bound protein 10; *H19*, Imprinted maternally expressed non-coding transcript; *Igf2*, Insulin-like growth factor 2; *Igf2r*, Insulin-like growth factor 2 receptor; *KCNQ1OT1*, *KCNQ1*-overlapping transcript 1; *MEST*, Mesoderm-specific transcript, mouse, homolog of; *Peg10*, paternally expressed 10; *RHOX*, Reproductive homeobox X-linked; *SNRPN*, Small nuclear ribonucleoprotein polypeptide N; *ZAC*, Zac tumor suppression gene (also known as *PLAGL1*, pleomorphic adenoma gene-like 1).

**Table 3 ijms-21-01728-t003:** Sperm proteins predicted to play a role in fertilization and preimplantation embryo development by Gene Ontology annotations and Mouse Genome Informatics databases.

Process	Proteins
*Fertilization* Acrosome reaction Binding of sperm to zona pellucida (ZP) Penetration of the ZP Sperm‒oocyte plasma membrane fusion Oocyte activation	AAAS, ABHD2, ACR, ADAM2, ADAM20, ADAM21, ADAM30, AKAP3, AKAP4, APOB, ASH1L, ATP1A4, ATP8B3, B4GALT1, BAX, BCL2L1, BSPH1, CATSPER1, CATSPER2, CATSPER4, CATSPERB, CATSPERD, CATSPERG, CCDC136, CCT2, CCT3, CCT4, CCT5, CCT7, CCT8, CD46, CD9, CDK1, CLGN, CLIC4, CRISP1, DEFB126, DNALI1, DUOX2, ELSPBP1, EQTN, GLIPR1L1, GLRB, GNPDA1, H3F3A, HEXB, HSPA1L, HVCN1, INSL6, IQCF1, IZUMO1, KCNU1, KLHL10, LY6K, MAEL, MFGE8, NLRP5, NOX5, PARK7, PCSK4, PKDREJ, PLCB1, PLCD4, PLCZ1, PRSS37, RNASE10, ROPN1B, SERPINA10, SERPINA5, SLC22A16, SMAD4, SMCP, SPA17, SPACA3, SPACA6, SPACA7, SPAG1, SPAG8, SPAM1, SPESP1, SPINK2, SPTBN4, STX2, SYCP2, TARBP2, TCP1, TCP11, TDRKH, TEKT3, TEX11, TEX15, TRIM36, TRPC7, TUBGCP3, UBAP2L, UBE3A, UBXN8, WBP2NL, ZAN, ZP1, ZP2, ZPBP, ZPBP2
*Preimplantation embryo development* First divisions Morula Blastocyst	AP1G1, AP2M1, ARFGEF2, ARHGDIB, ARPC3, ATP6V0C, B3GAT3, B3GNT5, BRCA2, BRK1, BSG, C1QBP, C2orf61, CALCA, CAPN2, CAPN5, CDH1, CDK11B, CENPF, CHD4, CTNNA1, CTTN, CUL3, DAD1, DDR1, DDX20, DMAP1, DSC3, EIF3B, EIF6, FAF1 FASN, FKBP4, HNRNPL, HSBP1, IGFBP7, KIF11, LATS1, MCL1, MMP2, MMP9, NASP, NDEL1, NEPRO, NUP214, ODF2, PCSK1, PCSK5, PCYT1A, PDCD2, PFN1, PPM1B, PRKRA, PRLR, PRPF19, PSMC3, PSMC4, PTGS2, PUM1, RACGAP1, RBM19, RPL7L1, RPS19, RPSA, RRM1, RUVBL1, SBDS, SCGB1A1, SEC24D, SMURF2, SNAP23, SOD1, SPP1, SRSF3, TBP, TGFBR1, TGFBR2, TIMM23, TIMP1, TKT, TMED10, TOP1, TPM3, TRIM28, TRRAP, TUBG1, VCP, ZPR1

**Legend:** AAAS, Achalasia‒Addisonianism‒Alacrimia syndrome; ABHD2, abhydrolase domain containing 2, acylglycerol lipase; ACR, acrosin; ADAM2, ADAM metallopeptidase domain; AKAP, A-kinase anchoring protein 3; APOB, apolipoprotein B; AP1G1, adaptor related protein complex 1 subunit gamma 1; AP2M1, adaptor related protein complex 2 subunit mu 1; ARFGEF2, ADP ribosylation factor guanine nucleotide exchange factor 2; ARHGDIB, Rho GDP dissociation inhibitor beta; ARPC3, actin related protein 2/3 complex subunit 3; ASH1L, ASH1 like histone lysine methyltransferase; ATP1A4, ATPase Na+/K+ transporting subunit alpha 4; ATP8B3, ATPase phospholipid transporting 8B3; ATP6V0C, ATPase H+ transporting V0 subunit c; B3GAT3, beta-1,3-glucuronyltransferase 3; B3GNT5, UDP-GlcNAc:betaGal beta-1,3-N-acetylglucosaminyltransferase 5; B4GALT1, beta-1,4-galactosyltransferase 1; BAX, BCL2-associated X, apoptosis regulator; BCL2L1, BCL2 like 1; BRCA2, BRCA2 DNA repair associated; BRK1, BRICK1 subunit of SCAR/WAVE actin nucleating complex; BSG, basigin (Ok blood group); BSPH1, binder of sperm protein homolog 1; C1QBP, complement C1q binding protein; C2orf61, sperm-tail PG-rich repeat containing 4; CALCA, calcitonin related polypeptide alpha; CAPN, calpain; CDH1, cadherin 1; CATSPER, cation channel sperm associated 1; CCDC136, coiled-coil domain containing 136; CCT, chaperonin containing TCP1 subunit; CDK1, cyclin dependent kinase 1; CDK11B, cyclin dependent kinase 11B; CENPF, centromere protein F; CHD4, chromodomain helicase DNA binding protein 4; CLGN, calmegin; CLIC4, chloride intracellular channel 4; CRISP1, cysteine rich secretory protein 1; CTNNA1, catenin alpha 1; CTTN, cortactin; CUL3, cullin 3; DAD1, defender against cell death 1; DDR1, discoidin domain receptor tyrosine kinase 1; DDX20, DEAD-box helicase 20; DEFB126, defensin beta 126; DMAP1, DNA methyltransferase 1-associated protein 1; DNALI1, dynein axonemal light intermediate chain 1; DSC3, desmocollin 3; DUOX2, dual oxidase 2; EIF3B, eukaryotic translation initiation factor 3 subunit B; EIF6, eukaryotic translation initiation factor 6; ELSPBP1, epididymal sperm binding protein 1; EQTN, equatorin; GLIPR1L1, GLIPR1 like 1; GLRB, glycine receptor beta; GNPDA1, glucosamine-6-phosphate deaminase 1; FAF1, Fas-associated factor 1; FASN, fatty acid synthase; FKBP4, FKBP prolyl isomerase 4; H3F3A, H3.3 histone A; HEXB, hexosaminidase subunit beta; HNRNPL, heterogeneous nuclear ribonucleoprotein L; HSBP1, heat shock factor binding protein 1; HSPA1L, heat shock protein family A (Hsp70) member 1 like; HVCN1, hydrogen voltage gated channel 1; IGFBP7, insulin like growth factor binding protein 7; INSL6, insulin like 6; IQCF1, IQ motif containing F1; IZUMO1, izumo sperm‒egg fusion 1; KCNU1, potassium calcium-activated channel subfamily U member 1; KIF11, kinesin family member 11; KLHL10, kelch like family member 10; LATS1, large tumor suppressor kinase 1; LY6K, lymphocyte antigen 6 family member K; MAEL, maelstrom spermatogenic transposon silencer; MCL1, MCL1 apoptosis regulator, BCL2 family member; MFGE8, milk fat globule EGF and factor V/VIII domain containing; MMP, matrix metallopeptidase; NASP, nuclear autoantigenic sperm protein; NDEL1, nudE neurodevelopment protein 1 like 1; NEPRO, nucleolus and neural progenitor protein; NLRP5, NLR family pyrin domain containing 5; NOX5, NADPH oxidase 5; NUP214, nucleoporin 214; ODF2, outer dense fiber of sperm tails 2; PARK7, Parkinsonism-associated deglycase; PCSK, proprotein convertase subtilisin/kexin; PCSK4, proprotein convertase subtilisin/kexin type 4; PKDREJ, polycystin family receptor for egg jelly; PLCB1, phospholipase C beta 1; PLCD4, phospholipase C delta 4; PLCZ1, phospholipase C zeta 1; PCYT1A, phosphate cytidylyltransferase 1, choline, alpha; PDCD2, programmed cell death 2; PFN1, profilin 1; PPM1B, protein phosphatase, Mg2+/Mn2+ dependent 1B; PRSS37, serine protease 37; PRKRA, protein activator of interferon-induced protein kinase EIF2AK2; PRLR, prolactin receptor; PRPF19, pre-mRNA processing factor 19; PSMC, proteasome 26S subunit, ATPase; PTGS2, prostaglandin-endoperoxide synthase 2; PUM1, pumilio RNA binding family member 1; RACGAP1, Rac GTPase activating protein 1; RBM19, RNA binding motif protein 19; RNASE10, ribonuclease A family member 10 (inactive); ROPN1B, rhophilin-associated tail protein 1B; RPL7L1, ribosomal protein L7 like 1; RPS19, ribosomal protein S19; RPSA, ribosomal protein SA; RRM1, ribonucleotide reductase catalytic subunit M1; RUVBL1, RuvB like AAA ATPase 1; SBDS, SBDS ribosome maturation factor; SCGB1A1, secretoglobin family 1A member 1; SEC24D, SEC24 homolog D, COPII coat complex component; SLC22A16, solute carrier family 22 member 16; SMAD4, SMAD family member 4; SMCP, sperm mitochondria-associated cysteine rich protein; SMURF2, SMAD specific E3 ubiquitin protein ligase 2; SNAP23, synaptosome-associated protein 23; SOD1, superoxide dismutase 1; SPA17, sperm autoantigenic protein 17; SPACA, sperm acrosome-associated 3; SPAG, sperm-associated antigen; SPAM1, sperm adhesion molecule 1; SPESP1, sperm equatorial segment protein 1; SPINK2, serine peptidase inhibitor Kazal type 2; SPP1, secreted phosphoprotein 1; SPTBN4, spectrin beta, non-erythrocytic 4; SRSF3, serine and arginine rich splicing factor 3; STX2, syntaxin 2; SYCP2, synaptonemal complex protein 2; TARBP2, TARBP2 subunit of RISC loading complex; TBP, TATA-box binding protein; TCP, t-complex; TDRKH, tudor and KH domain containing; TEKT3, tektin 3; TEX, testis expressed; TGFBR, transforming growth factor beta receptor; TIMM23, translocase of inner mitochondrial membrane 23; TIMP1, TIMP metallopeptidase inhibitor 1; TKT, transketolase; TMED10, transmembrane p24 trafficking protein 10; TOP1, DNA topoisomerase I; TPM3, tropomyosin 3; TRRAP, transformation/transcription domain-associated protein; TRIM28, tripartite motif containing 28; TRIM36, tripartite motif containing 36; TRPC7, transient receptor potential cation channel subfamily C member 7; TUBG1, tubulin gamma 1; TUBGCP3, tubulin gamma complex-associated protein 3; UBAP2L, ubiquitin-associated protein 2 like; UBE3A, ubiquitin protein ligase E3A; UBXN8, UBX domain protein 8; VCP, valosin-containing protein; WBP2NL, WBP2 N-terminal like; ZAN, zonadhesin (gene/pseudogene); ZP, zona pellucida glycoprotein; ZPBP, zona pellucida binding protein; ZPR1, ZPR1 zinc finger.

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
