# Peer review of "Molecular Biology of Spermatogenesis: Novel Targets of Apparently Idiopathic Male Infertility"

_ijms, 2020, doi:10.3390/ijms21051728_

Round 1
Reviewer 1 Report
In the manuscript “Molecular biology of spermatogenesis: novel targets of apparently idiopathic male infertility”, the authors proposed to discuss molecular mechanisms related to spermatogenetic failure and the role of sperm-carried molecular factors involved in oocyte fertilization and embryo growth. The paper is timely and as some interest although the authors are encouraged to include more in-depth discussion. The paper is mostly descriptive and lacks some relevant issues that must be included. I have some specific comments that may be useful.
Specific comments:
- In their “omics” analysis, the authors perhaps should at least briefly discuss sperm metabolome and the relevance for its physiology. There are several studies focused on those topics. In fact, authors should also highlight that spermatogenesis is a process highly dependent of metabolic regulation and because of that, sensitive to metabolic cues. That can help to sustain the hypothesis.
- I would expect an in-depth analysis of future perspectives on the subject and the limitations. The authors are encouraged to further discuss their view on the knowledge and the state of the literature so far. What can we expect for the near future? What is the expected clinical impact of these omics? As it is the paper is mostly descriptive and lacks In-depth discussion and critical view on the future of the topic.
Author Response
Comment 1: In their “omics” analysis, the authors perhaps should at least briefly discuss sperm metabolome and the relevance for its physiology. There are several studies focused on those topics. In fact, authors should also highlight that spermatogenesis is a process highly dependent of metabolic regulation and because of that, sensitive to metabolic cues. That can help to sustain the hypothesis.
Answer to comment 1: Done as requested. Please see lines 353-371.
Comment 2: I would expect an in-depth analysis of future perspectives on the subject and the limitations. The authors are encouraged to further discuss their view on the knowledge and the state of the literature so far. What can we expect for the near future? What is the expected clinical impact of these omics? As it is the paper is mostly descriptive and lacks In-depth discussion and critical view on the future of the topic.
Answer to comment 2: Thank you for this comment. We have added such considerations in lines 373-394.
Reviewer 2 Report
Dear authors,
This manuscript reviews the biology of spermatogenesis and the influence of this proccess on the fertility. This is a elegant and well written review which extensively addresses the molecular factors involved in human fertility. The topic developed is considered of great interest for the development of therapeutic targets in the study of idiopathic male infertility.
Author Response
Comment 1: Dear authors, this manuscript reviews the biology of spermatogenesis and the influence of this process on fertility. This is an elegant and well-written review which extensively addresses the molecular factors involved in human fertility. The topic developed is considered of great interest for the development of therapeutic targets in the study of idiopathic male infertility.
Answer to comment 1: Thank you for your comment. We appreciated the time you spent to review this manuscript.
Round 2
Reviewer 1 Report
The revised version of the manuscript has improved.